# Greatwall promotes cell transformation by hyperactivating AKT in human malignancies

**Jorge Vera[1], Lydia Lartigue[2], Suzanne Vigneron[1], Gilles Gadea[1], Veronique Gire[1], Maguy Del Rio[3], Isabelle Soubeyran[2], Frederic Chibon[2], Thierry Lorca[1]*, Anna Castro[1]***

[1]Centre de Recherche de Biochimie Macromoléculaire, Université de Montpellier, Montpellier, France; [2]Department of Medical Oncology, Institut Bergonié, Institut National de la Santé et de la Recherche Medicale, Université Bordeaux Segalen, Bordeaux, France; [3]Institut de Recherche en Cancérologie de Montpellier, Université de Montpellier, Montpellier, France

**Abstract** The PP2A phosphatase is often inactivated in cancer and is considered as a tumour suppressor. A new pathway controlling PP2A activity in mitosis has been recently described. This pathway includes the Greatwall (GWL) kinase and its substrates endosulfines. At mitotic entry, GWL is activated and phosphorylates endosulfines that then bind and inhibit PP2A. We analysed whether GWL overexpression could participate in cancer development. We show that GWL overexpression promotes cell transformation and increases invasive capacities of cells through hyperphosphorylation of the oncogenic kinase AKT. Interestingly, AKT hyperphosphorylation induced by GWL is independent of endosulfines. Rather, GWL induces GSK3 kinase dephosphorylation in its inhibitory sites and subsequent SCF-dependent degradation of the PHLPP phosphatase responsible for AKT dephosphorylation. In line with its oncogenic activity, we find that GWL is often overexpressed in human colorectal tumoral tissues. Thus, GWL is a human oncoprotein that promotes the hyperactivation of AKT via the degradation of its phosphatase, PHLPP, in human malignancies.


***For correspondence:** thierry.lorca@crbm.cnrs.fr (TL); anna.castro@crbm.cnrs.fr (AC)

**Competing interests:** The authors declare that no competing interests exist.






## Introduction

The Greatwall (GWL) kinase was originally identified in *Drosophila* where it was first proposed to be involved in the control of mitotic progression (**Bettencourt-Dias et al., 2004**; **Yu, 2004**). Biochemical experiments in *Xenopus* egg extracts demonstrated that during mitosis GWL is required to inhibit the protein phosphatase 2A complexed to B55 regulatory subunit (PP2AB55), a phosphatase that dephosphorylates cyclinB-cyclin-dependent kinase 1 (CDK1) substrates (**Castilho et al., 2009**; **Vigneron et al., 2009**). However, PP2AB55 inhibition by GWL is not direct, but through phosphorylation of the two endosulfines ARPP19 and ENSA that once phosphorylated bind and inhibit PP2AB55 (**Gharbi-Ayachi et al., 2010**; **Mochida et al., 2010**). The mammalian orthologue of GWL, originally named Microtubule-Associated Serine Threonine Kinase Like (MASTL), is also involved in the control of mitotic division. *MASTL* silencing in human cells and knockout in mice increase PP2AB55 activation and decrease phosphorylation of cyclinB-CDK1 substrates leading to important mitotic defects (**Alvarez-Fernandez et al., 2013**; **Burgess et al., 2010**). GWL kinase activity is tightly regulated during mitotic division by phosphorylation at the C terminus and the T-loop domains, possibly by cyclinB-CDK1 and the *Xenopus* orthologue of the Polo-like kinase (PLX1) (**Blake-**

**eLife digest** In order to form a tumour, cancer cells have to overcome the controls that normally prevent cells from dividing too often. These controls include an enzyme called PP2A, which inhibits cell division by regulating the activity of other proteins. In a normal cell, PP2A may be temporarily deactivated from time to time to enable the cell to divide. However, PP2A is permanently deactivated in many cancer cells, which allows these cells to divide many times in quick succession.

A protein called Greatwall is involved in deactivating PP2A in healthy cells, but it was not clear whether increases in Greatwall activity can promote the formation of tumours. Here, Vera et al. use a variety of cell biology techniques to address this question. The experiments show that increasing the amount of Greatwall in human and mouse cells can promote some of the transformations needed for these cells to become cancerous. Also, Greatwall increases the growth of tumours in mice. These effects are caused by the over-activation of a protein called AKT, which is already known to promote the formation of many cancers.

Vera et al. show that Greatwall regulates AKT activity using a different pathway to how it influences PP2A activity. Further experiments revealed that many human tumours, especially those from patients with colon cancer, produce excessive amounts of Greatwall protein. Together, these findings show that Greatwall can promote the development of cancer. A future challenge is to understand how this works and to find out whether high levels of Greatwall are a common feature of other human cancers.


***Hodek et al., 2012***; ***Vigneron et al., 2011***). Unlike the regulation of its kinase activity, nothing is known about the mechanisms controlling GWL protein levels.

PP2A is one of the main serine-threonine phosphatases involved in the control of multiple cellular signalling pathways in mammalian cells. This holoenzyme comprises three subunits: a catalytic subunit (PP2AC, or C subunit), a scaffolding subunit (PP2AA, or A subunit) and a regulatory subunit (PP2AB, or B subunit) that is responsible for substrate specificity. This assembly complexity is crucial for PP2A large substrate repertoire and wide diversity of physiological functions (***Janssens et al., 2008***; ***Virshup and Shenolikar, 2009***). Several PP2A holoenzymes are considered to be tumour suppressors and are functionally inactivated in cancer. Loss of activity of distinct PP2A holocomplexes mediates oncogenesis by activating different signalling pathways, including the kinases AKT and mitotic-activated protein kinase (MAPK) (***Andrabi et al., 2007***; ***Rodriguez-Viciana et al., 2006***). Particularly, PP2AB55 deregulation has been observed in breast, prostate, and colon cancers. Moreover, deletions in *PPP2R2A* (gene encoding B55$\alpha$ isoform) are frequently detected in prostate and breast tumours (***Cheng et al., 2011***; ***Curtis et al., 2012***) and the promoter silencing of *PPP2R2B* (gene encoding B55$\beta$ isoform) has been found in colorectal cancer (***Yasutis et al., 2010***).

Several oncogenic pathways are regulated by B55. The B55$\alpha$ subunit participates in the regulation of the RAS-RAF-MAPK signalling pathway (***Ory et al., 2003***) and controls MAPK signalling via direct dephosphorylation of the inhibitory phosphorylation site ($Ser^{259}$) of RAF1 (***Adams et al., 2005***). In FL5.12 pro-lymphoid cells, PP2AB55$\alpha$ directly associates with AKT and promotes dephosphorylation of AKT-activating residue ($Thr^{308}$) (***Kuo et al., 2008***). B55$\beta$ binds to phosphoinositide-dependent kinase 1 (PDK1) and modulates its activity towards MYC phosphorylation (***Tan et al., 2010***). Finally, B55$\gamma$ can negatively regulate c-Src activity through dephosphorylation of Ser12, a residue required for c-Jun N-terminal *Kinase* (JNK) activation by c-Src (***Eichhorn et al., 2007***).

As GWL-dependent phosphorylation of ARPP19 and ENSA promotes their binding to and inhibition of PP2AB55, we analysed whether GWL participates in cell transformation and cancer development through inhibition of PP2AB55 tumour suppressor activity.

## Results

### GWL overexpression promotes transformation of immortalised mammary gland cells and primary human fibroblast

We asked whether GWL overexpression could promote transformation of immortalised non-transformed mammary gland cells. To this aim, we stably overexpressed pMXs-based constructs encoding wild type (WT), hyperactive kinase (K72M) or kinase dead (G44S) GWL or with empty vector (CT) into MCF10A cells, and we compared their proliferative and transforming capacities to those observed in MCF10A overexpressing the V12Ras oncogene (***Figure 1A***). Stably overexpressed V12Ras and WT and K72M GWL forms are shown in ***Figure 1A***. The levels of these three ectopic proteins are similar, although we often observed a lower expression of the hyperactive form of GWL. Increased expression of WT and K72M GWL forms significantly raised cell proliferation (***Figure 1A***), reduced cell contact inhibition (***Figure 1B***) and promoted anchorage independent-cell growth (***Figure 1C***) compared to control cells (CT), although in a lesser extent than V12Ras overexpression. Interestingly, overexpression of G44S GWL induced cell death probably by acting as a dominant negative since when the levels of WT GWL were increased concomitantly with those of a G44S GWL no effect on cell viability was observed (***Figure 1—figure supplement 1***). Similar results were obtained in NIH3T3. In these cell lines, kinase dead GWL expression was lethal, whereas overexpression of WT or hyperactive GWL strongly promoted cell proliferation, invasion and migration and decreased cell—cell contact inhibition (***Figure 1—figure supplement 2***). This indicates that GWL overexpression might favour cell transformation in addition to promoting cell proliferation, migration, and invasion and reducing contact inhibition.

We thus checked whether GWL could also induce cell transformation in primary human fibroblasts. Overexpression of WT GWL did not overtly affect the cell phenotype. Conversely, overexpression of K72M GWL led to accumulation of cells with a double DNA content (***Figure 1D***, FACs profile) and cell senescence (***Figure 1D***, β-galactosidase staining), a characteristic feature of activated oncogene overexpression in cultured primary cells (***Lowe et al., 2004***). As it has been reported that human fibroblast transformation by some oncogenes, such as Ras (***Land et al., 1983***), requires cell immortalisation and inhibition of the p53 pathway, we then overexpressed WT and K72M GWL in human fibroblasts immortalised by overexpression of telomerase reverse transcriptase (hTERT) and SV40 large T antigen (***Figure 1E***). As expected, accumulation of cells with double DNA content arrest was no longer observed in K72M GWL overexpressing cells. Moreover, proliferation of fibroblasts expressing WT or K72M GWL strongly increased (***Figure 1E***) to levels comparable to those induced by the V12-Ras oncogene and anchorage-independent growth was promoted (***Figure 1F***), demonstrating that GWL has transforming capacity.

### GWL overexpression is involved in the invasive behaviour of colon and mammary tumour cell lines

Based on these results, we first decided to investigate the role of GWL in the aggressive behaviour of breast cell lines. We chose MDA-MB-231 and MDA-MB-231-D3H2LN (D3H2LN) breast cancer cell lines. These cells present similar genetic background (D3H2LN coming from MDA-MB-231 cells) but different metastatic potential (D3H2LN: high metastatic clone of MDA-MB-231 selected by subcutaneous mouse xenograft). As expected, D3H2LN cells presented increased cell proliferation, migration, and invasion than MDA-MB-231 (***Figure 2A***, CT versus D3H2LN). Interestingly, the former cells also displayed higher GWL expression levels (***Figure 2A*** upper panels). To test whether the different invasive behaviour of these cell lines could be mediated by GWL overexpression, we stably overexpressed WT, K72M, and G44S GWL in MDA-MB-231 cells (displaying low endogenous GWL levels and metastatic capacity) and knocked down GWL by shRNA in D3H2LN cells (presenting high endogenous GWL levels and metastatic capacity). Overexpression of WT and particularly K72M GWL in MDA-MB-231 cells significantly increased cell proliferation, invasion, and migration to reach similar levels than D3H2LN cells (***Figure 2A***). Conversely, unlike its negative effect on cell viability in NIH3T3 and MCF10A cells, overexpression of G44S GWL did not affect cell behaviour in MDA-MB-231 cells probably due to the fact that they display higher endogenous GWL levels.

When GWL was knocked down in D3H2LN cells, cell proliferation, migration, and invasion dramatically dropped to levels close or even lower than those observed in MDA-MB-231 cells

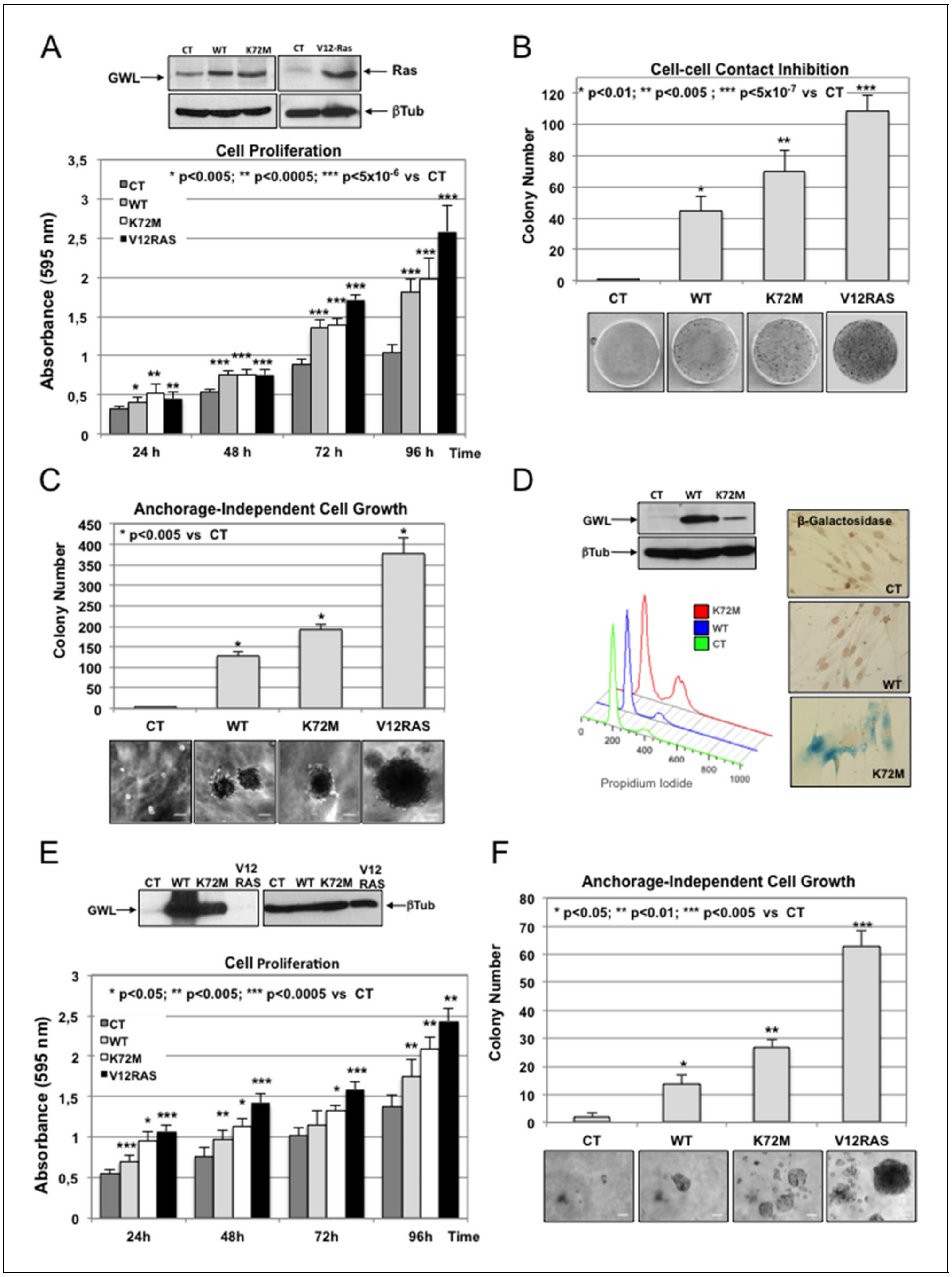


**Figure 1.** GWL overexpression promotes cell transformation in immortalised mammary gland cells and in primary human fibroblasts. (**A-C**) MCF10A cells were stably infected with either empty pMXs vector (CT) or plasmids coding for wild type (WT) or hyperactive (K72M) GWL. A MCF10A stable clone overexpressing the V12RAS oncogene was used as a positive control. Cell proliferation (**A**), cell—cell contact inhibition (**B**) and anchorage-independent cell growth (**C**) were assayed in these cells. Colonies stained by crystal violet in normal seeded plates (cell—cell contact inhibition) or in soft-agar plates (anchorage-independent cell growth) were counted and results presented in a graph. Results are shown as the mean of three different experiments ± SD. Two-tailed unpaired Student *t* tests were performed to determine statistical relevance; significant p values are shown. Scale bars, 60 μm. (**D**) Primary human fibroblasts were infected with empty pMXs vector (CT) or with plasmids coding for the indicated GWL proteins. FACS profile and β-Galactosidase staining (blue) of these cells are shown. (**E**,**F**) Human fibroblasts expressing hTERT and SV40 T large antigen were infected with an empty vector (CT) or with plasmids encoding WT or K72M GWL. An hTERT-SV40-V12Ras cell line was used as a positive control. Cell proliferation (**E**) and anchorage-independent cell growth (**F**) are shown. Results are the mean ± SD of three different experiments.

*Figure 1 continued*

Two-tailed unpaired Student's *t* tests were performed to determine the statistical relevance; significant p values are shown. Scale bars, 100 µm. SD, standard deviation.


The following figure supplements are available for figure 1:

**Figure supplement 1.** G44S kinase dead form of GWL in MCF10A cell lines could act as dominant-negative of endogenous GWL.


**Figure supplement 2.** NIH3T3 cells were stably infected with either the pMXs empty vector (CT) or the GWL wild type (WT) or hyperactive mutant form (K72M).


(***Figure 2B***). Decrease of these oncogenic properties induced by *GWL* shRNA was specific of GWL knockdown because this effect was rescued in SH GWL-stable cell lines when they were transfected with a plasmid encoding an shRNA-resistant WT GWL but not with the same plasmid coding for the kinase dead form (***Figure 2—figure supplement 1***).

We observed similar results when the different forms of GWL where overexpressed or knocked down in SW480 and SW620, respectively, two colorectal cancer cells also displaying similar genetic background but different invasive properties (***Figure 2—figure supplement 2***).

These results suggest that GWL mediates their different oncogenic properties via GWL expression level and subsequent increased kinase activity. Accordingly, GWL activity was significantly higher in WT and K72M GWL overexpressed cells than in CT cells or in cells stably overexpressing G44S GWL (***Figure 2—figure supplement 3***).

## GWL overexpression promotes tumour growth in vivo

To investigate whether GWL overexpression/hyperactivation influences tumour growth in vivo, we injected MDA-MB-231 cells stably transfected with empty vector (CT) or WT GWL or D3H2LN cells that stably express control shRNA (SH LUC) or *GWL* shRNA (SH GWL) in athymic nude mice and monitored tumour growth regularly. Tumours derived from MDA-MB-231-GWL cells grew significantly faster than those derived from MDA-MB-231 control cells. Similarly, tumours derived from D3H2LN-*GWL*-shRNA cells grew significantly slower than those obtained from D3H2LN-LUC-shRNA cells, showing that GWL has a positive effect on tumour cell proliferation also in vivo (***Figure 3A***).

## GWL oncogenic activities are mediated through AKT phosphorylation at S473

To identify the signalling pathway(s) involved in GWL oncogenic activities, we simultaneously analysed the relative site-specific phosphorylation of 43 kinases involved in different oncogenic pathways in extracts from SW480 and MDA-MB-231 cells that overexpress WT GWL or an empty pMXs vector (CT) and from SW620 and D3H2LN cells stably transfected with control (SH LUC) or *GWL* shRNA (SH GWL) by using a HumanPhospho-kinase Antibody Array (see 'Materials and methods'). Our results show a decrease of the phosphorylation of the inhibitory sites of glycogen synthase kinase-3 (GSK3) (S9/S21) in both GWL overexpressing MDA-MB-231 and SW480 cells as well as an increase of the phosphorylation of AKT on S473. Moreover, AKT phosphorylation on S473 was significantly reduced in *GWL* knockdown D3H2LN and SW620 cells (***Figure 3B***). No difference in AKT phosphorylation at T308 was observed (***Figure 3—figure supplement 1***). We confirmed the effect of GWL on AKT phosphorylation at S473 by Western blot analysis in these cells and in NIH3T3, HeLa, MCF10A cells, and primary human fibroblasts. AKT phosphorylation on S473 was significantly increased in all these cell lines when WT or K72M GWL were overexpressed (***Figure 3—figure supplement 2***), whereas no effect was observed when the kinase dead form was used. Conversely, S473 phosphorylation was significantly reduced in SW620 and D3H2LN cells that stably express the *GWL* shRNA compared to cells expressing SH LUC. These results suggest that GWL overexpression could promote the oncogenic properties of these cells by increasing AKT phosphorylation at S473.

To test this hypothesis, we investigated the effect of the AKT inhibitor VIII on proliferation, migration, and invasion of MDA-MB-231 cells and on anchorage-independent cell growth of MCF10A cells. First, by performing a dose-response analysis, we found that 1 µM was the highest dose of

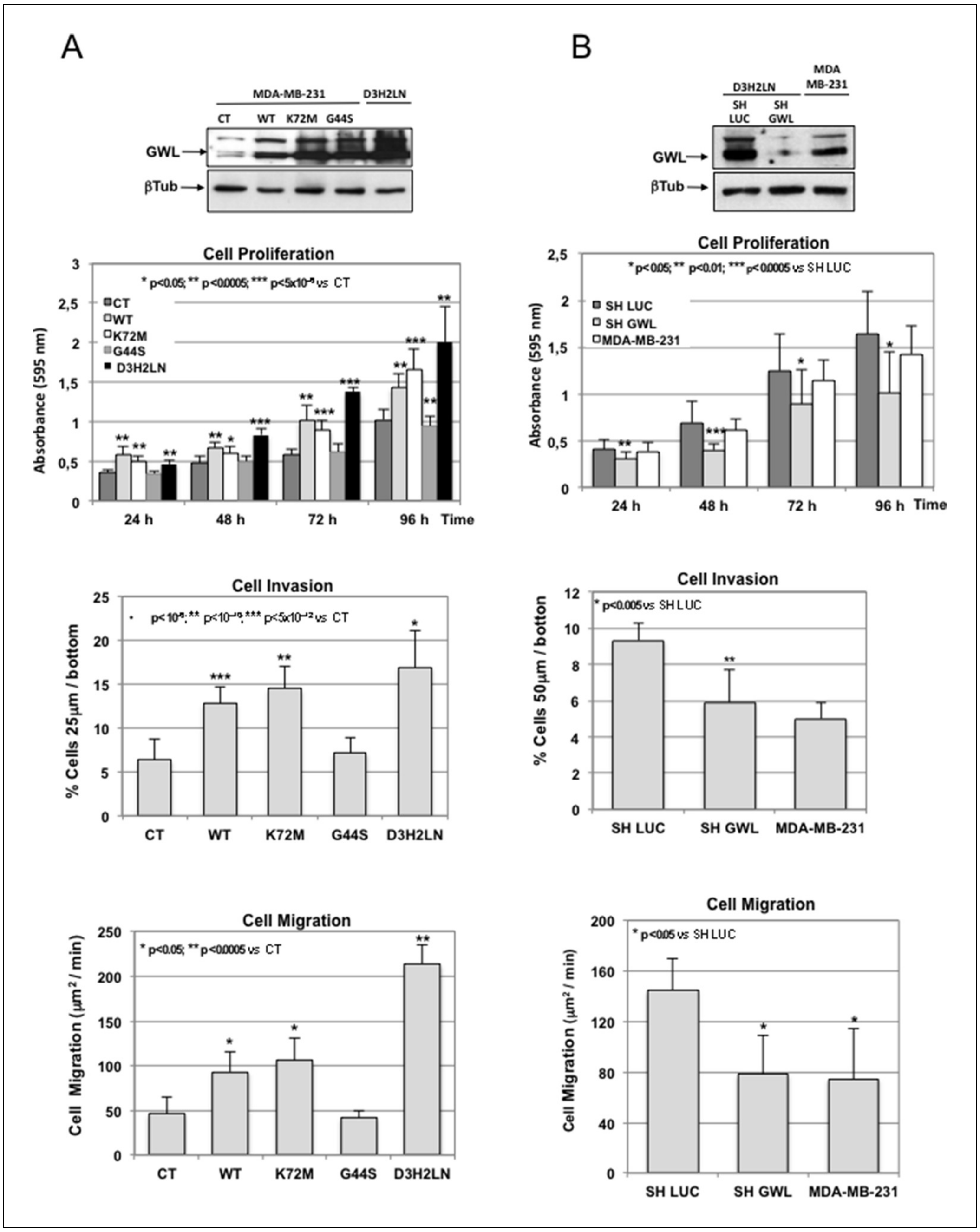


**Figure 2.** GWL overexpression is involved in the invasive behaviour of mammary tumour cell lines. (**A**) MDA-MB-231 cells were infected with empty pMXs vector (CT) or with plasmids coding for the indicated GWL proteins. Cell proliferation, invasion, and migration were measured in these cells and in D3H2LN cells and represented as mean ± SD. Two-tailed unpaired Student's *t* tests were performed to determine the statistical relevance; significant p values are shown. (**B**) D3H2LN cells were stably infected with control (SH LUC) or anti-*GWL* shRNAs. Cell proliferation, invasion, and migration were then measured in these cells, compared to those of MDA-MB-231 cells and represented as mean ± SD. Two-tailed unpaired Student's *t* tests were performed to determine the statistical relevance; significant p values are shown. GWL,Greatwall; SD, standard deviation.


The following figure supplements are available for figure 2:

**Figure supplement 1.** Decrease of oncogenic properties induced in D3H2LN cells by *GWL* shRNA was rescued by co-transfection with an shRNA-resistant WT GWL but not with the GWL kinase dead form.


*Figure 2 continued*

**Figure supplement 2.** GWL overexpression is involved in the invasive behaviour of colon tumour cell lines.

**Figure supplement 3.** GWL activity is higher in WT and K72M GWL overexpressed cells than in control (CT) cells or in cells stably overexpressing G44S GWL.

inhibitor that did not significantly affect cell proliferation in parental MDA-MB-231 (***Figure 4—figure supplement 1***). Then, we assessed the effect of this dose on phosphorylation on S473 of AKT, cell proliferation, invasion, migration, and anchorage-independent growth in cells overexpressing WT GWL or an empty pMXs vector (CT) (***Figure 4A–D***). As expected, 1 µM of AKT inhibitor did not have any effect in MDA-MB231 control cells. Conversely, in GWL overexpressing cells AKT phosphorylation was reduced and cell proliferation, invasion, and migration significantly decreased to levels close or equal to those observed in controls. Moreover, inhibitor VIII also strongly reduced the capacity of GWL-overexpressing MCF10A cells to grow in soft agar. We obtained similar results in SW480 (***Figure 4—figure supplement 2***) and confirmed them with another AKT inhibitor (MK2206) in MDA-MB-231 and MCF10A cells (***Figure 4—figure supplement 3***). However, we did not observe any effect on S473 AKT phosphorylation or cell proliferation in MDA-MB-231 GWL overexpressed cells when other kinase controlling cell proliferation such as MEK was inhibited (***Figure 4—figure supplement 4***).

Altogether, these findings demonstrate that AKT inhibition is sufficient to reverse the phenotype induced by GWL overexpression in SW480, MDA-MB231, and MCF10A cells, suggesting that GWL oncogenic effects are mostly mediated by increased AKT phosphorylation on S473.

## AKT phosphorylation on S473 is not due to direct phosphorylation by GWL, or increased mTORC2 activity or decreased PP2AB55/PP2AB56 activity

AKT phosphorylation on S473 in GWL-overexpressing cells could be due to direct phosphorylation by GWL, to increased activity of mTORC2, the kinase responsible for this phosphorylation, or to inhibition of the phosphatase(s) promoting AKT dephosphorylation. First, we used in vitro phosphorylation assays to rule out direct AKT phosphorylation by GWL (***Figure 5—figure supplement 1***). We then checked whether mTORC2 activity was required to maintain AKT phosphorylation on S473 by knocking down Rictor, an essential component of the mTORC2 complex, in MDA-MB-231 cells. Although AKT phosphorylation at S473 was reduced by about 50% in control MDA-MB-231 cells, upon Rictor knockdown, no change in AKT phosphorylation was observed in GWL-overexpressing cells (***Figure 5A***). Similarly, Rictor knockdown led to a significant reduction of cell proliferation in control MDA-MB-231 cells, whereas no effect was observed in GWL-overexpressing cells. These results indicate that increased AKT phosphorylation at S473 is not due to higher mTORC2 kinase activity or to direct phosphorylation by GWL.

We then asked whether it could be the result of the inhibition of the phosphatase responsible for AKT dephosphorylation. Previous results from our laboratory showed that GWL inhibits PP2AB55 through phosphorylation of its inhibitors ARPP19 and ENSA at mitotic entry (***Gharbi-Ayachi et al., 2010***). We thus tested whether GWL induced AKT phosphorylation at S473 through ARPP19/ENSA-dependent inhibition of PP2AB55 by stably overexpressing ARPP19 and ENSA (WT and the respective S62A and S67A mutants that cannot be phosphorylated by GWL) in MCF10A cells (***Figure 5B***). Cell proliferation was increased following WT ARPP19 overexpression, although significantly less than following overexpression of WT or K72M GWL. Conversely, proliferation was not affected by overexpression of S62A ARPP19, indicating that increased proliferation induced by WT ARPP19 overexpression is dependent on the binding and inhibition of PP2AB55. No change in cell proliferation was observed upon overexpression of WT or S67A ENSA. Moreover, none of these proteins had any effect on AKT phosphorylation at S473, supporting the hypothesis that AKT activation upon GWL overexpression is not mediated by ARPP19/ENSA-dependent PP2AB55 inhibition.

We finally tested the putative role of PP2AB55 and PP2AB56 in GWL-induced AKT phosphorylation at S473 by transiently expressing B55 or B56 in control or GWL-overexpressing MDA-MB-231 cells (***Figure 5C***). Overexpression of B55 and B56 promoted a significant drop of cell proliferation in

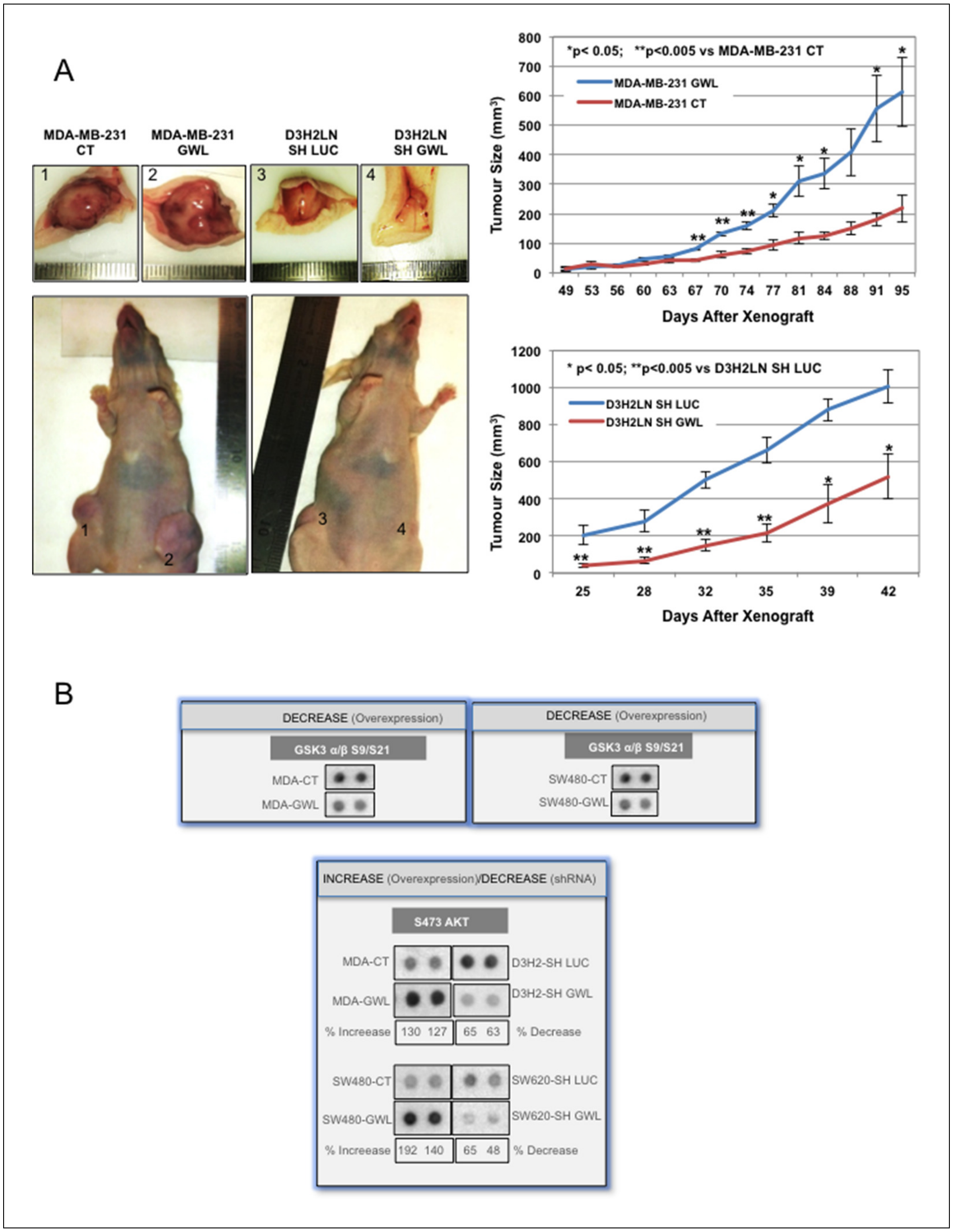


**Figure 3.** GWL overexpression promotes tumour growth in vivo and increases AKT phosphorylation on S473. (**A**) Two representative animals with tumours induced by subcutaneous injection of MDA-MB-231 cells stably infected with the pMXs empty vector (CT) or this plasmid encoding wild type GWL (GWL) (left animal, tumours 1 and 2, respectively) or D3H2LN cells that stably express SH LUC or SH GWL (right animal, tumours 3 and 4, respectively). Photographs show animals and tumours at the end of the experiment. Tumour size (mean ± SEM) was measured at different time points after xenograft in both groups of animals (n=4). Two-tailed unpaired Student's *t* tests were performed to determine the statistical relevance; significant p values are shown. (**B**) The indicated cell clones were lysed and equal amounts of total proteins were incubated with the human phospho-kinase array. GSK3 α/β phosphorylation at S9/S21 in the different cell lines are shown. AKT phosphorylation at S473 correlated with the GWL expression level in all tested cell lysates. Shown is the percent of increase of S473 phosphorylation signal calculated by densitometry using the ImageJ software in GWL overexpressing cells compared to CT or in GWL knockdown cells compared SH LUC cells. GWL, Greatwall.


both control and GWL-overexpressing MDA-MB-231 cells without decreasing AKT phosphorylation on S473. This indicates that inhibition of these two phosphatases is not involved in GWL-induced AKT phosphorylation on S473.

## AKT phosphorylation on S473 induced by GWL overexpression is due to a decrease of PHLPP protein levels

PHLPP is the phosphatase responsible for AKT dephosphorylation on S473 (***Gao et al., 2005***; ***Brognard et al., 2007***). As our data suggest that induction of AKT phosphorylation on S473 by GWL overexpression is the result of decreased dephosphorylation of this site, we tested whether GWL modulate PHLPP protein levels. First, we evaluated endogenous PHLPP in MDA-MB-231, MCF10A and SW480 cells that overexpress WT or the indicated mutants of GWL, ARPP19 or ENSA and in D3H2LN and SW620 treated with control (shLUC) or a GWL shRNA (***Figure 6A***). PHLPP levels were significantly decreased in MDA-MB-231 cells and were completely undetectable in MCF10A and SW480 cells overexpressing WT or K72M GWL, whereas no effect on the endogenous levels of this phosphatase was observed when either WT or mutant ARPP19 and ENSA were overexpressed. On the contrary, the knockdown of GWL promoted an increase of the levels of this phosphatase. Analysis of PHLPP protein levels in GWL-overexpressing or control MCF10A cells transiently transfected with PHLPP at 0, 2, 4, 6, 8, and 10 hr after addition of cycloheximide indicated that GWL overexpression strongly promoted PHLPP proteolysis (***Figure 6B***). It has been previously reported that PHLPP levels are regulated by a GSK3-dependent phosphorylation of this protein in four different sites followed by SKP1-Cullin-Fbox-βTrCP (SCF-βTrCP)-mediated ubiquitination and proteosomal degradation (***Li et al., 2009***). In order to check whether PHLPP proteolysis induced in GWL overexpressed cells is proteasome dependent, we added the proteasome inhibitor MG132 to the culture media. Under these conditions, PHLPP was accumulated in both CT and GWL overexpressing MCF10A cells, although the accumulation observed in CT cells was far below the one induced in GWL overexpressing cells (***Figure 6C***). We next checked whether this degradation is ubiquitin-dependent. To answer this question we overexpressed a mutant of PHLPP in which the four sites (S847, T851, S867 and S869) phosphorylated by GSK3 were substituted by alanines (PHLPP–S/T4A). This mutant cannot be ubiquitinated by the SCF-βTrCP and is not degraded (***Li et al., 2009***). As shown in ***Figure 6D***, PHLPP–S/T4A levels were maintained in both CT and GWL-overexpressing MCF10A cells treated with cycloheximide indicating that GWL overexpression promotes SCF-βTrCP-dependent ubiquitination and proteasome-dependent degradation of PHLPP.

We next asked whether GWL could directly phosphorylate PHLPP and promote its ubiquitination and degradation, but we did not observe any direct phosphorylation of PHLPP by GWL (***Figure 6—figure supplement 1***). As phosphorylation of the inhibitory sites of GSK3 is reduced in GWL-overexpressing MDA-MB-231, SW480, and MCF10A cells and increased in D3H2LN and SW620 GWL knockdown cells (***Figure 3B*** and ***Figure 6E***), we tested whether GSK3 inhibition could affect PHLPP levels in GWL-overexpressing cells. As expected, PHLPP protein level was reduced in GWL-overexpressing cells compared with control cells (***Figure 6F***, GWL and CT, 0 μM). However, addition of a GSK3 inhibitor (from 2.5 to 10 μM) increased in a dose-dependent manner PHLPP levels in GWL-overexpressing cells but not in control cells. This suggests that PHLPP level reduction in GWL-overexpressing cells is the consequence of increased GSK3 activity that leads to PHLPP ubiquitination and degradation.

Finally, if AKT hyperactivation in GWL-overexpressing cells is the result of a loss of PHLPP, the ectopic overexpression of this phosphatase should reduce proliferation, invasion and anchorage-independent cell growth in GWL overexpressing cells to levels close to those observed in control cells. ***Figure 7A*** show the PHLPP levels present in CT and GWL MCF10A overexpressed cells 24 hr after transient overexpression. PHLPP ectopic amount left at the different time-points of the proliferation assay is also shown (***Figure 7B***). According to a role of PHLPP degradation in AKT hyperactivation in GWL overexpressing cells, the overexpression of this phosphatase significantly decreased cell proliferation and invasion in both GWL-overexpressing and control MCF10A cells (***Figure 7B and 7C***) and differences in these properties between these cell lines completely disappeared. We obtained the same results when MDA-MB-231 cells were used (***Figure 7—figure supplement 1***). Finally, overexpression of PHLPP in MCF10A that stably express GWL also reduced their capacity to grow in soft-agar (***Figure 7D***). These data indicate that GWL overexpression promotes AKT

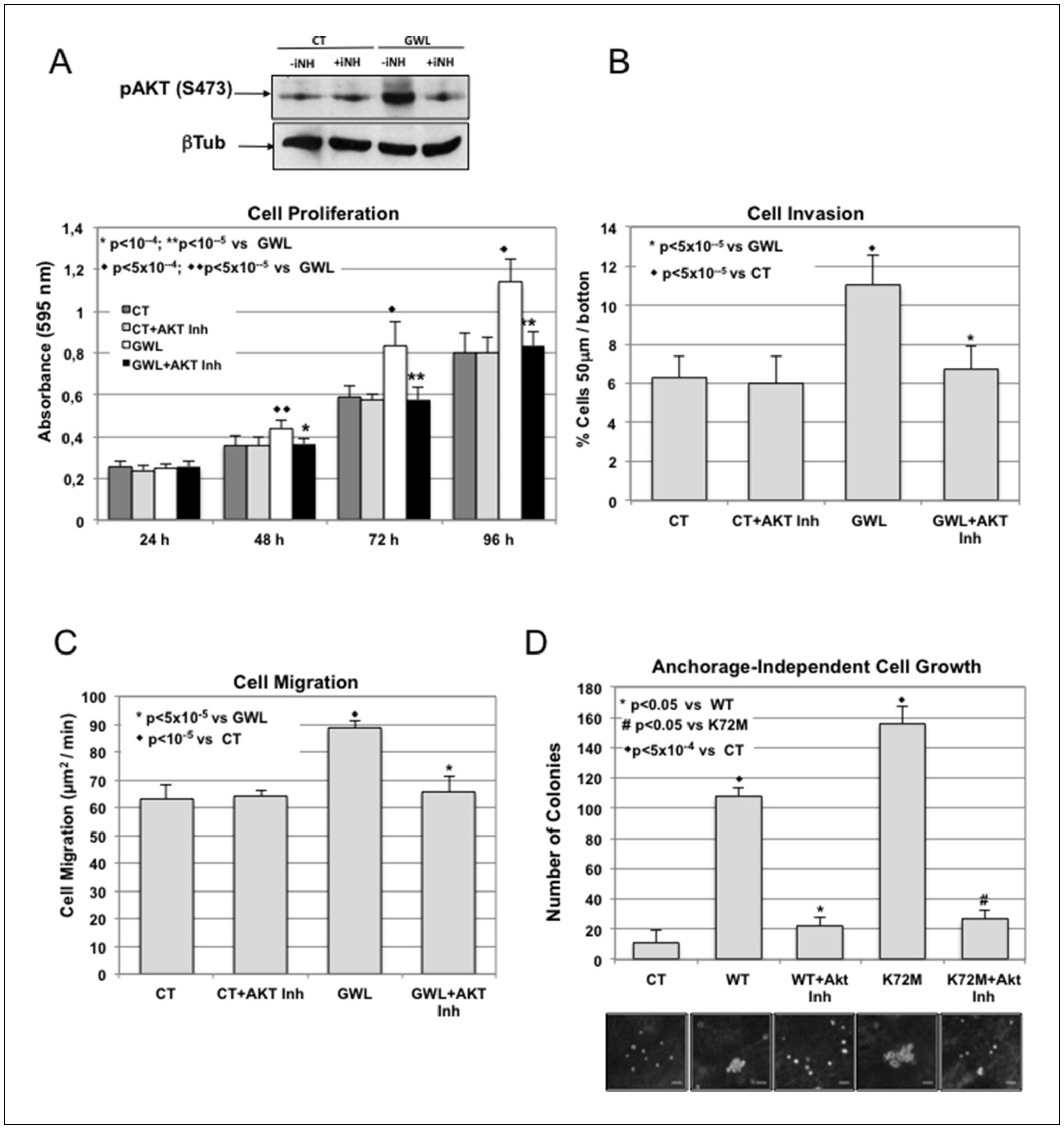


**Figure 4.** GWL activation promotes increased cell proliferation, migration, and invasion through AKT phosphorylation at S473. (**A-C**) MDA-MB-231 cells that stably overexpress GWL or not (CT) were plated and 4 hr later supplemented with a medium containing 1 µM of the AKT Inhibitor VIII. Medium containing AKT inhibitor was replaced every 2 days. Phosphorylation of AKT on S473 (24 hr after inhibitory treatment) and cell proliferation, invasion, and migration were analysed in these cells. (**D**) MCF10A cells that overexpress WT or K72M GWL were grown in the presence of 1 µM of the AKT Inhibitor VIII. As for (**A–C**) medium containing AKT inhibitor was replaced every 2 days. The capacity of these cells to grow in soft-agar was analysed by quantifying the number of colonies formed. Results are the mean ± SD of three different experiments. Two-tailed unpaired Student's *t* tests were performed to determine the statistical relevance; significant p values are shown. Scale bars, 60 µm. GWL, Greatwall; SD, standard deviation.

The following figure supplements are available for figure 4:

**Figure supplement 1.** Inhibitors VIII dose response curves in MDA-MB-231. MDA-MB-231 cells were submitted to increasing doses of the AKT inhibitor VIII and cell proliferation was then measured at 24, 48, 72, and 96 hr. AKT inhibitor was replaced at 48 hr after addition.

**Figure supplement 2.** Inhibitors VIII dose response curves in SW480 control cells and effect of 1 µM of the AKT Inhibitor VIII in cell proliferation, invasion and migration.

**Figure supplement 3.** The inhibitor of AKT, MK2206, like Inhibitor VIII, rescues GWL phenotype.

**Figure supplement 4.** The MEK inhibitor U0126 has no effect in GWL overexpressed MDA-MB-231 cell proliferation.

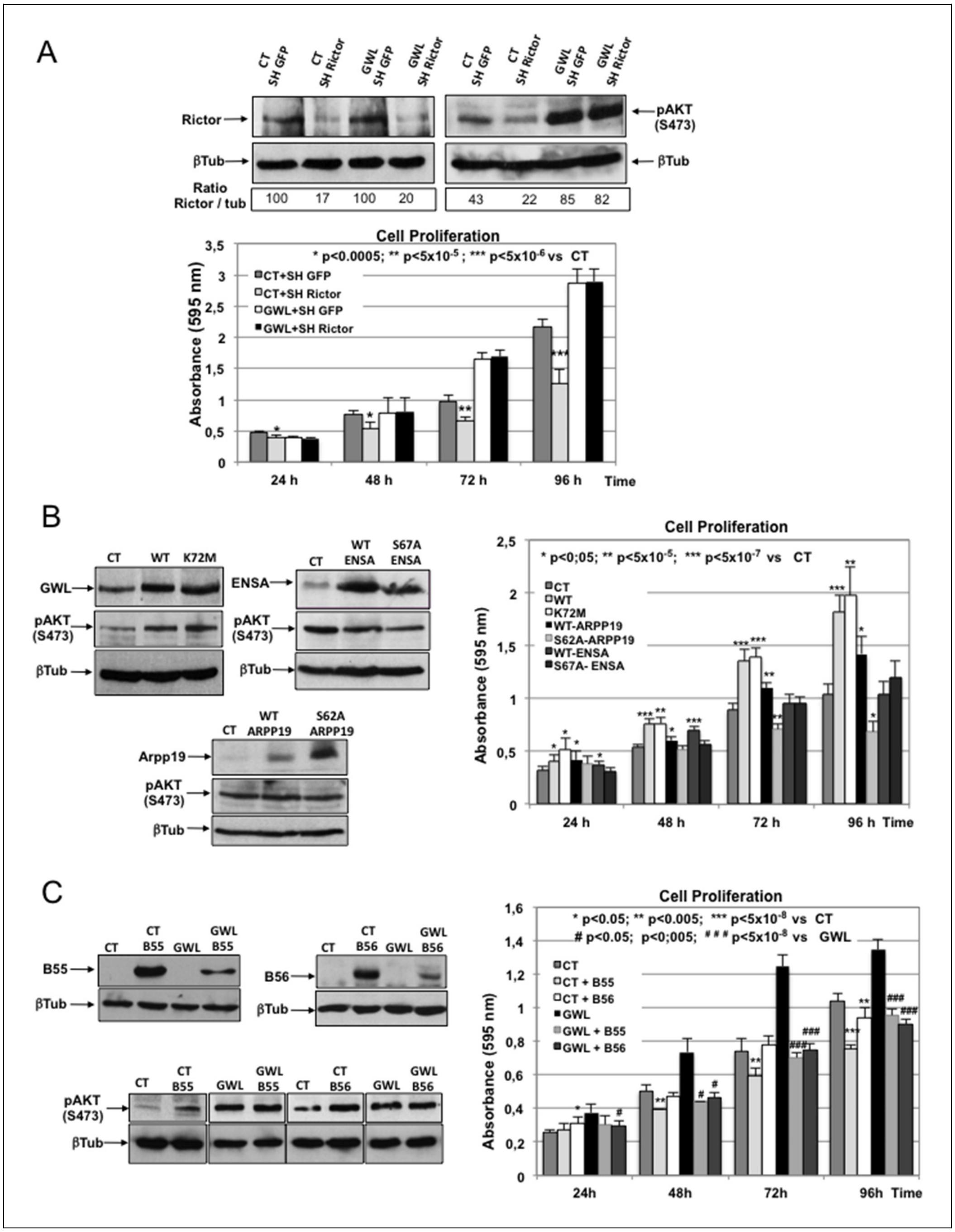


**Figure 5.** AKT phosphorylation on S473 induced by GWL overexpression is neither due to increased mTORC2 activity nor to decreased PP2AB55 activity through ARPP19/ENSA phosphorylation. (**A**) Control or GWL-overexpressing MDA-MB-231 cells were stably infected with control (SH GFP) or *Rictor* shRNAs (SH Rictor) and the levels of Rictor and AKT phosphorylation at S473 were determined by Western blotting. The Rictor / β-tubulin ratio and AKT phosphorylation at S473 / β-tubulin ratio were calculated by densitometry using the ImageJ software. Cell proliferation in these cells was also measured. (**B**) MCF10A cells were infected with viruses encoding the indicated proteins and the expression levels of ARPP19, ENSA, β-tubulin, and AKT phosphorylated at S473 were determined. Cell proliferation in the different cell lines was also assessed. Results are the mean ± SD of three different experiments. Two-tailed unpaired Student's *t* tests were performed to determine the statistical relevance; significant p values are shown. (**C**) MDA-MB-231 cells that stably express GWL (GWL) or not (CT) were transfected with B55 or B56 and the levels of these two proteins and of phosphorylated AKT at S473 were analysed by Western blotting. Cell proliferation was then measured and results shown in a bar graph (mean ± SD). Two-tailed

*Figure 5 continued*

unpaired Student's *t* tests were performed to determine the statistical relevance; significant p values are shown. GWL, Greatwall; SD, standard deviation.


The following figure supplement is available for figure 5:

**Figure supplement 1.** GWL does not directly phosphorylate AKT.


hyperphosphorylation on S473 that in turns induces an increase in cell proliferation, invasion, and anchorage-independent cell growth by decreasing PHLPP levels.

### GWL is overexpressed in human malignancies

Altogether, these results demonstrate that GWL overexpression promotes cell transformation and invasion in vitro and tumour growth in vivo. However, it is not known whether GWL is overexpressed in human transformed cell lines or in tumours. We thus checked GWL expression levels by Western blotting in 5 non-transformed and 20 transformed cell lines from 8 different tissues (kidney, cervix, skin, colon, muscle, prostate, breast, and white blood cells). In nineteen of the twenty transformed cell lines, GWL levels were three- to ten-fold higher than in non-transformed cells from the same tissue type (***Figure 8A***). Similarly, in crude extracts from 13 sets of matched non-tumour (NT) and tumour tissue (TT) biopsies of patients with stage IV colorectal adenocarcinoma GWL expression levels were significantly higher in 9 TT than in the matched NT samples (***Figure 8B***).

To further expand these observations, we analysed GWL expression levels by immunostaining using a tissue microarray (TMA) made of matched NT and TT biopsies from 76 patients with colorectal cancer (CRC) (***Soubeyran et al., 2011***). GWL was frequently overexpressed in TT samples compared to the matched NT samples (***Figure 8C***). Quantitative analyses confirmed this result, revealing that GWL median intensity fluorescence was increased by 25.6% in the TT samples compared to the NT controls (***Figure 8D***). We then examined GWL expression at different stages of tumour progression. To this aim, patients' tumours were categorised in stage I/II, III or IV groups and GWL expression level evaluated in the NT and TT samples of each group by measuring the median fluorescent intensity of the green signal (***Figure 8E***). GWL was equally overexpressed at all tumour stages, suggesting a direct link between GWL overexpression and oncogenic transformation. Finally, we determined the number of patients with high GWL expression in TT and found that 56% of patients (42 from 75) presented at least a 20% increase in GWL levels compared to controls (***Figure 8F***). These results indicate that GWL overexpression is frequent at all CRC stages, suggesting a direct role of this kinase in cell transformation.

## Discussion

As GWL regulates the activity of the tumour suppressor PP2A by phosphorylating its inhibitors ARPP19 and ENSA, we investigated whether GWL overexpression could participate in the oncogenic process. Our results demonstrate that GWL can promote cell transformation, proliferation and migration/invasion, but independently from the ARPP19/ENSA-PP2AB55 pathway, the only known GWL pathway.

We show that GWL overexpression promotes cell transformation in two different immortalised cell lines (NIH3T3 and MCF10A). Moreover, like most oncogenes, GWL overexpression induced the senescence checkpoint in primary human fibroblasts and promoted cell proliferation and anchorage-independent cell growth in human fibroblast with high levels of hTERT and large-T antigen of SV40. Besides this cell transforming effect, GWL also promoted the aggressive behaviour of cancer cells. Specifically, GWL increased in a dose-dependent manner the migratory and invasive properties of two colon and two breast cancer cell lines with a similar genetic background. Accordingly, *GWL* knockdown in more aggressive breast and colon cell lines (D3H2LN and SW620 cells, respectively) decreased cell proliferation, migration and invasion to levels comparable to those observed in less metastatic cell lines (MDA-MB-231 and SW480). The opposite effect was observed when GWL was overexpressed in the less metastatic cell lines. Finally, we show that GWL levels modulate tumour growth in vivo as well (three-fold increase of tumour size in tumours derived from cancer cells

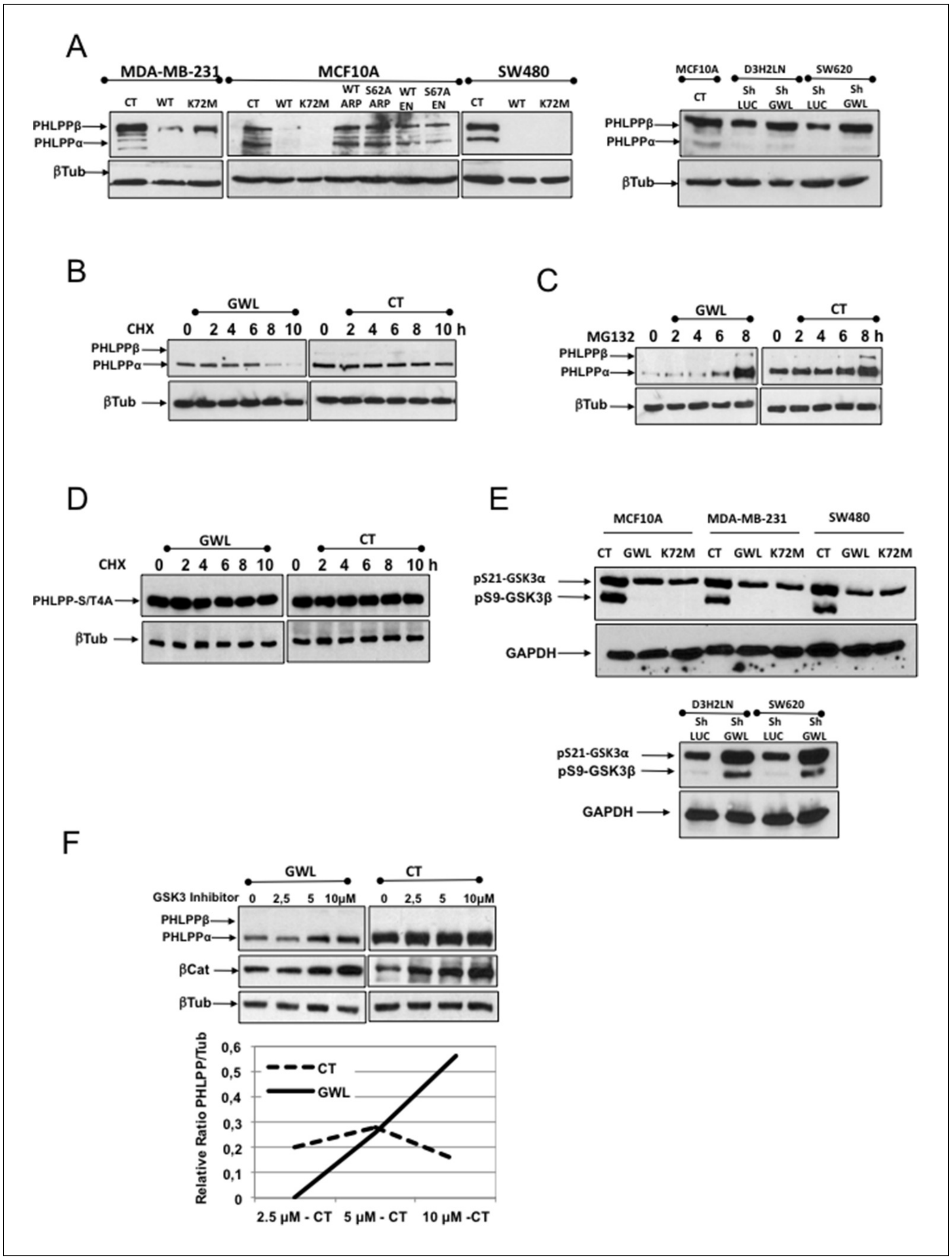


**Figure 6.** AKT phosphorylation at S473 in GWL-overexpressing cells is due to increased ubiquitin/proteasome-dependent degradation of PHLPP. (**A**) The levels of endogenous PHLPP and β-tubulin (loading control) were assessed by Western blotting in the indicated cell lines overexpressing GWL (WT and K72M) or not (CT) or the WT or mutant forms of ARPP19 or ENSA and in D3H2LN and SW620 cells treated with a control or a GWL shRNA. (**B**) MCF10A cells that overexpress GWL or the empty vector pMXs (CT) were transiently transfected with PHLPP and 24 hr later supplemented with 40 μg/ml of cycloheximide (CHX) and recovered at 0, 2, 4, 6, 8, or 10 hr after CHX treatment to check PHLPP levels by Western blotting using anti-PHLPP antibodies. (**C**) As for (**B**) except that the proteasome inhibitor MG132 (25 μM) instead of CHX was used. (**D**) MCF10A cells overexpressing (GWL) or not (CT) GWL were transiently transfected with the PHLPP-S/T4A mutant. After 24 hr, cells were supplemented with CHX (40 μg/ml) and the levels of PHLPP-S4A measured at 0, 2, 4, 6, 8, or 10 hr after CHX addition by Western blot by using anti-PHLPP antibodies. (**E**) Phosphorylation of the inhibitory sites S21 (GSK3α) and S9 (GSK3β) was analysed by Western blotting in control and MCF10A, MDA-MB-231, and SW480 cells that overexpress WT or K72M GWL or not (CT) and in D3H2LN and SW620 treated with a control (LUC) or a GWL shRNA. (**F**) MCF10A cells

*Figure 6 continued*

that stably overexpress GWL or not (CT) were transiently transfected with a plasmid-encoding PHLPP and 24 hr later incubated with 2.5, 5, or 10 μM of the GSK3 inhibitor SB216763 (SIGMA) for 5 hr. Samples were then recovered to measure PHLPP, βCatenin, and β-tubulin levels by Western blotting. Represented are the differences between the PHLPP/β-tubulin ratios of non-treated (CT) and inhibitor-treated cells (at the different concentrations).

The following figure supplement is available for figure 6:

**Figure supplement 1.** GWL does not directly phosphorylate PHLPP.

overexpressing GWL and two-fold decrease in tumours obtained from cancer cells in which *GWL* was silenced compared to controls).

Our study also reveals that the effect of GWL overexpression on cell proliferation, migration and invasion is mostly induced by AKT activation through the phosphorylation of its activating residue S473. Accordingly, S473 phosphorylation was observed in all analysed GWL overexpressing cell lines and AKT inhibition blunted the effects of GWL overexpression.

However, GWL does not phosphorylate S473 of AKT directly or indirectly by increasing mTORC2 kinase activity because Rictor knockdown had no effect on S473 phosphorylation level in GWL-overexpressing cells. These results indicate that phosphorylation of S473/AKT could be due to an inhibition of the phosphatase(s) responsible for its dephosphorylation. Our findings clearly demonstrate that AKT phosphorylation on S473 is not mediated by inhibition of PP2AB56 or of PP2AB55 through ARPP19/ENSA phosphorylation. Instead, they support the hypothesis that GWL overexpression stimulates the degradation of PHLPP, the phosphatase responsible for AKT dephosphorylation on S473 (***Gao et al., 2005***). PHLPP levels were significantly reduced in all GWL overexpressing cell lines due to increased ubiquitin and proteasome-dependent degradation. Moreover, PHLPP overexpression completely reversed the phenotype induced by GWL overexpression. To be targeted for proteosomal degradation, PHLPP is first phosphorylated by GSK3 and then ubiquitinated by SCF-βTrCP (***Li et al., 2009***). We observed a decrease of the phosphorylation levels of the GSK3 inhibitory residues S9/S21 upon GWL overexpression, suggesting that the lower PHLPP levels in GWL overexpressing cells are the result of GSK3 hyperactivation. Accordingly GSK3 inhibition induced a dramatic increase of PHLPP levels in GWL-overexpressing MCF10A cells, whereas no effect was observed in control cells.

This is the first work showing that GWL has additional roles besides ARPP19/ENSA phosphorylation and PP2AB55 inhibition. Our results suggest that GWL phosphorylates another protein(s) to promote GSK3 activation through dephosphorylation of its inhibitory sites (***Figure 9***). This would result in PHLPP ubiquitination and degradation and in the subsequent increase of AKT phosphorylation on S473 that confers to this protein robust transforming and metastatic capacities. However, the molecular mechanisms by which GWL promotes dephosphorylation of the inhibitory sites of GSK3 need to be further elucidated. In this line, it is known that AKT promotes phosphorylation of GSK3 in S9/S21 (***Li et al., 2009***), however, GWL overexpressed cells display a hyperactivated AKT but a S9/S21 dephosphorylated GSK3. One possible explanation for this counterintuitive physiological situation is that GWL would mediate dephosphorylation of the inhibitory sites of GSK3 by promoting the hyperactivation of the phosphatase responsible of this dephosphorylation. This would move the balance between phosphorylation of these sites by AKT and dephosphorylation by the phosphatase regulated by GWL towards dephosphorylated and activated GSK3.

GWL mutations have been described in cancer. Around 180 different mutations have been annotated for this protein (ICGC Data Portal; https://dcc.icgc.org). Most abundant mutations in cancer are frameshifts at positions K390 and K462 (4%), followed by missense mutations (S684L, S715L, S716L or G359D from 2–1%) although the latter ones are mostly distributed in the T-loop insert, in which changes in amino acid sequence do not affect GWL activity (***Vigneron et al., 2011***). Melanomas are the tumour pathologies in which GWL mutations have been most frequently detected (18%) followed by esophageal cancer (15%) and ovarian cancer (9%), however again, most GWL mutations detected in these cancers concern non-coding mutations or mutations at the T-loop fragment. No gain activity mutants have been described so far. Copy number amplification has also been described in breast cancer (8,6%) and in ovarian cancer (3,6%) (cBioPortal; http://www.cbioportal.

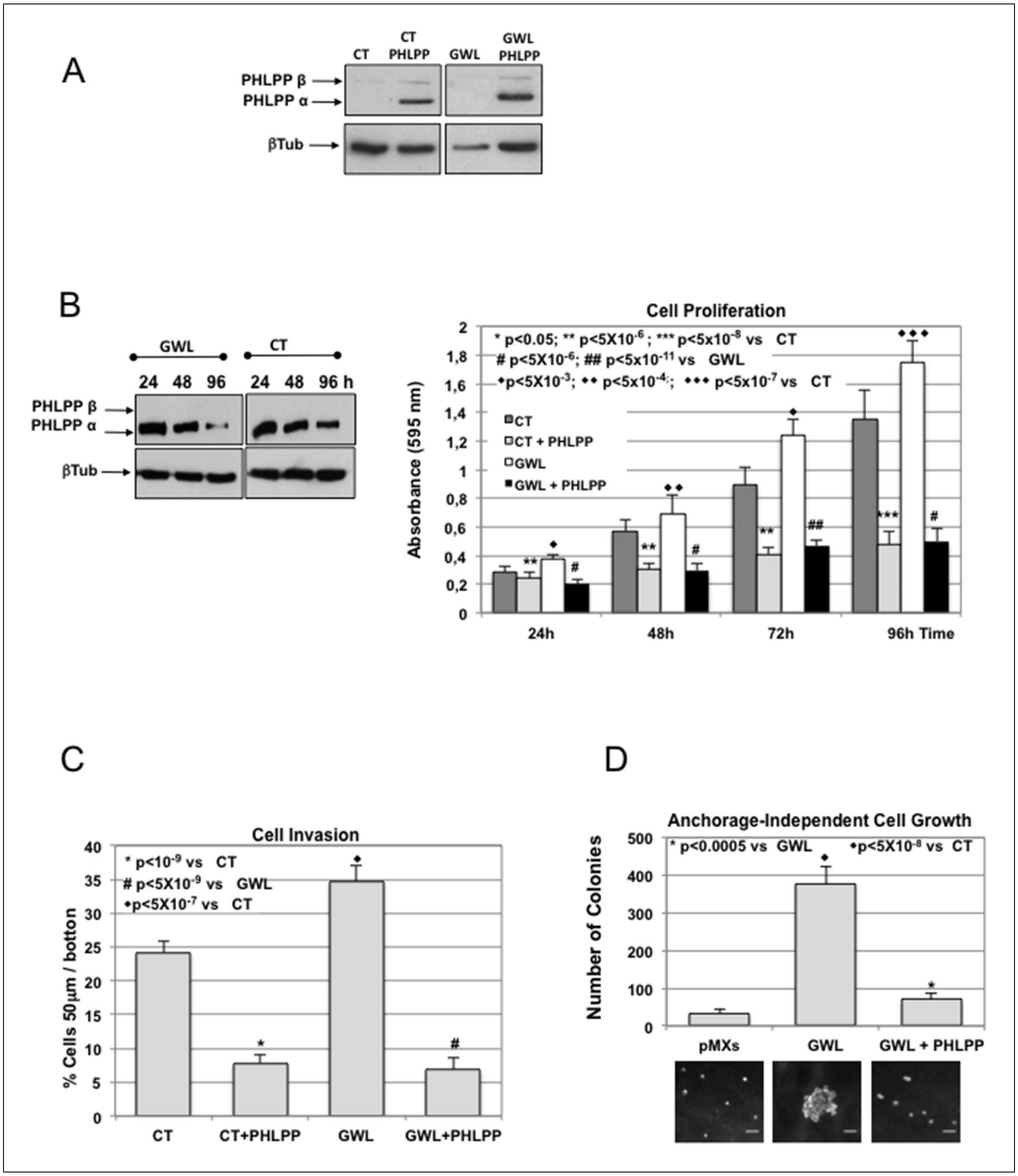


**Figure 7.** The oncogenic phenotype induced by GWL overexpression is rescued by the ectopic overexpression of PHLPP. (**A**) MCF10A cells that stably express or not GWL were transfected with a plasmid encoding PHLPP, and 24 hr later, the levels of this protein were measured and compared to CT cells. (**B**) 24 hr after transfection cells in (**A**) were seeded and cell proliferation was measured 24, 48, 72, and 96 hr later. The levels of PHLPP in these cells were measured at 24, 48, and 96 hr later. Results are shown in a bar graph as the mean ± SD of three different experiments. Two-tailed unpaired Student's *t* tests were performed to determine the statistical relevance; significant p values. (**C**,**D**) Cell invasion (**C**) and anchorage-independent cell growth (**D**) were measured in cells transfected in (**A**). Values were represented as the mean ± SD of three different experiments. Two-tailed unpaired Student's *t* tests were performed to determine the statistical relevance; significant p values are shown. Scale bars, 60 µm. CT, control; GWL, Greatwall; SD, standard deviation.


The following figure supplement is available for figure 7:

**Figure supplement 1.** Ectopic overexpression of PHLPP rescues GWL-induced phenotype in MDA-MB-231 cells.


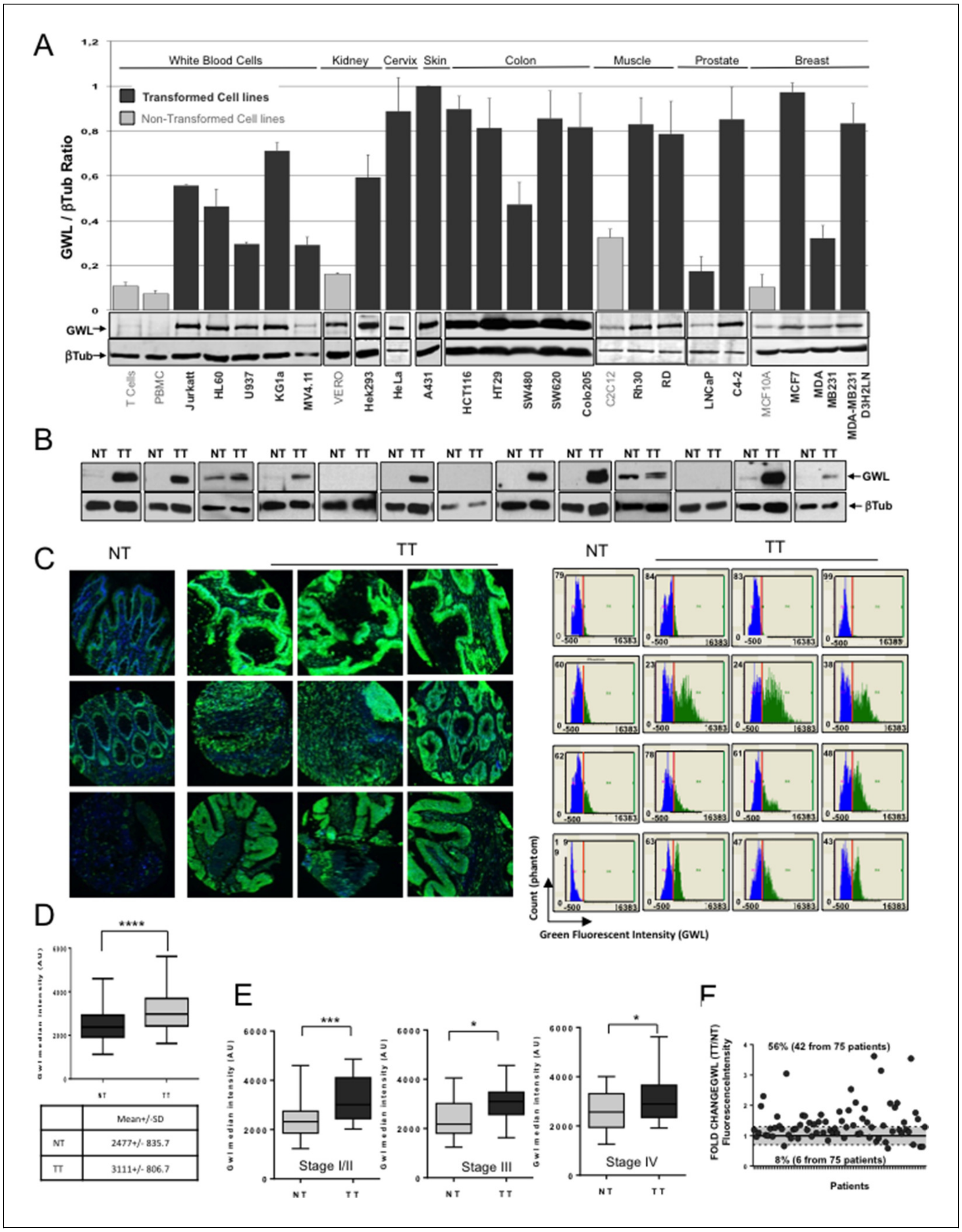


**Figure 8.** GWL overexpression is a common feature in transformed cell lines and in colorectal cancer. (**A**) GWL level in the indicated cell lines was assessed by Western blot analysis (lower panels) and the GWL / β-tubulin ratios shown in a bar graph (upper panel). (**B**) The expression levels of the GWL kinase and β-tubulin in crude extracts prepared from 13 sets of matched non-tumour (NT) and tumour (TT) samples from patients with stage IV primary colorectal adenocarcinomas were measured by Western blotting. (**C**) Representative images (left panels) and corresponding quantitative analysis (right panels) of GWL expression (green) in normal tissue (NT) and matching tumour (TT) after immunofluorescence staining of a colorectal TMA. The blue colour corresponds to DAPI staining. (**D**) Quantification of GWL expression in the NT and TT samples of (**C**). The median fluorescence intensity of the GWL signal was determined in the NT and TT samples of each patient and used to construct a box and whisker plot. GWL mean fluorescence intensity in each tissue is shown in the table (****$p<0.0001$ by paired t-test). (**E**) GWL median fluorescence intensity in the NT and corresponding TT of (**C**) according to the tumour stage (stage I/II: n=28, stage III: n=17 and stage IV: n=28 patients). ***$p<0.001$; *$p<0.05$ (paired t-test). (**F**) The fold change in GWL fluorescence intensity was calculated as the TT fluorescence intensity/NT fluorescence intensity ratio for each patient. The grey area represents the minimum threshold (20% increase/decrease) for considering GWL expression in TT as up-regulated or down-regulated compared to TT.


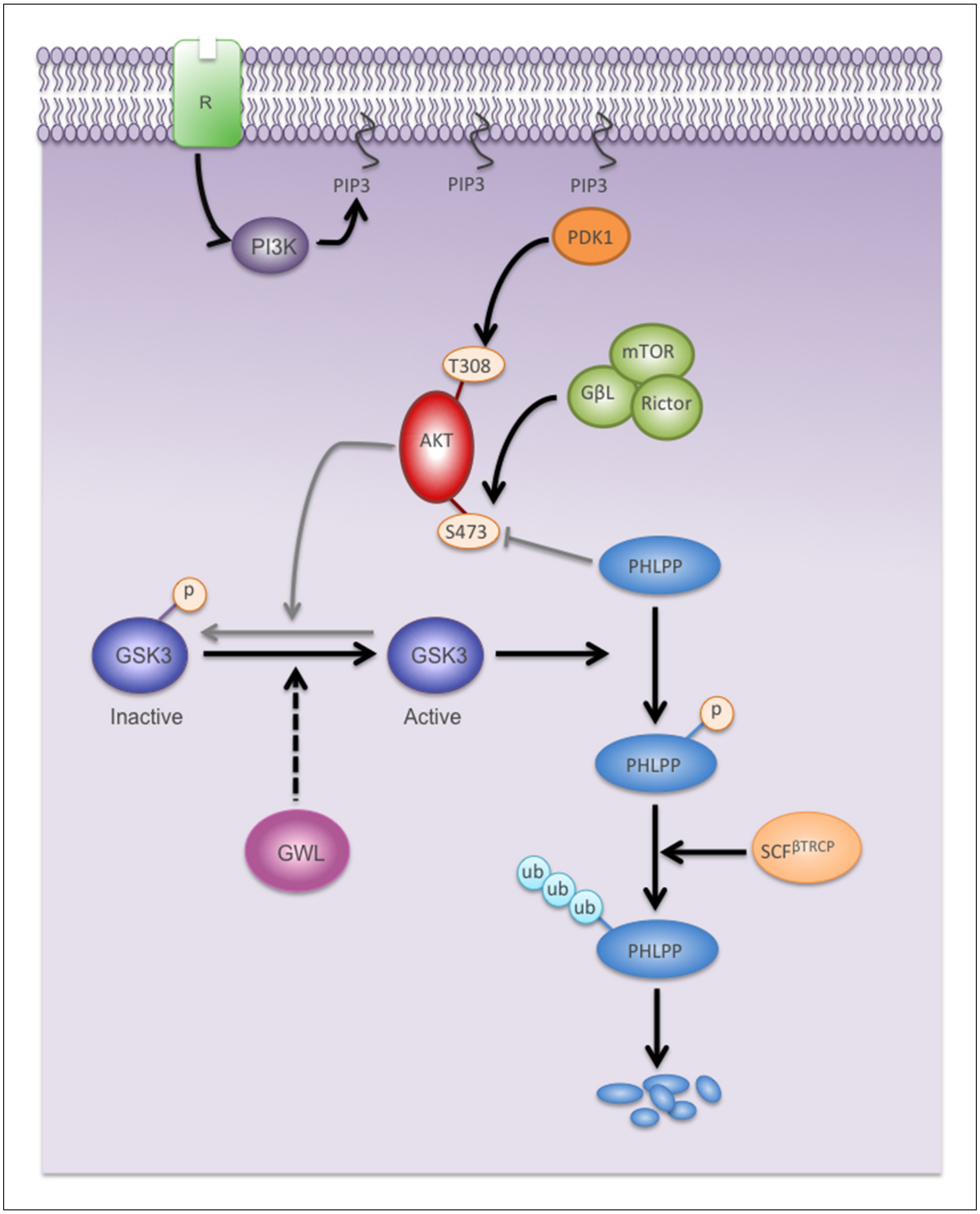


**Figure 9.** Model showing the mechanisms by which GWL overexpression promotes AKT hyperphosphorylation and hyperactivation in human cells. Black arrows represent pathways favoured by GWL overexpression.Grey arrows represent disfavoured pathways in GWL overexpressed cells. Dashed arrows represent indirect effect of GWL on GSK3 dephosphoryaltion and reactivation. R: Membrane receptor.

org). Finally, although increased gene expression has been identified in breast, colon, liver, testis, vagina and vulva cancers (GENT database; http://medical-genome.kribb.re.kr/GENT), no differences in protein expression have been identified so far (Human Protein Atlas; http://www.proteinatlas.org).

In this work, we show that GWL protein levels are increased in cancer. We observed increased GWL levels in most of the tested transformed cell lines and in 56% of the CRC samples analysed. Interestingly, GWL was overexpressed at all tumour stages, suggesting that GWL could be used as a marker of oncogenic transformation. Indeed, our results show that GWL overexpression is enough to

promote cell transformation in immortalised and in primary cells, a feature only displayed by powerful oncogenes, such as v-Ras. Besides its transforming capacity, GWL also confers robust invasive capacities to transformed cells. However, the finding that GWL levels are high from early tumour stages indicates that this kinase does not directly participate in metastasis formation/progression, but may instead facilitate the invasive process.

As far as we know GWL is the first oncogenic mitotic protein with transforming and invasive properties and the overexpression of which directly induces cell transformation in immortalised cells.

In summary, GWL expression level is increased in many transformed cells and in tumours at all stages. Moreover, it displays robust oncogenic and invasive capacities through a novel oncogenic signalling pathway that involves AKT, a major node in the crosstalk of tumour pathways. Therefore, GWL represents a good tumour marker candidate and also a potential target for future therapeutic approaches.

# Material and methods

## Cell culture

HEK-293t, SW480, SW620, MCF10A and MDA-MB-231 were purchased from ATCC collection (Middlesex, UK). Human Fibroblasts correspond to early passage fibroblasts isolated from neonatal human foreskin as described previously (**Baus, 2003**; **Dazard, 2000**). MDA-MB-231-D3H2LN was a Bioware Cell line purchased from Caliper LifeScience (Villepinte, France). Cells were regularly checked for mycoplasma contamination by using Mycoalert Detection Kit (#LT07-318) from Lonza (Levallois, France).

HEK-293t, SW480, SW620, MDA-MB-231 and D3H2LN were grown in DMEM supplemented with 10% FBS, 1 mM sodium pyruvate and antibiotics/glutamine (Invitrogen, Thermo Fisher, Illkirch, France). MCF10A cells were cultured in DMEM/F12 with 5% horse serum (Invitrogen, 1 mM sodium pyruvate, EGF (20 ng/ml, PEPROTECH, Neuilly-Sur-Seine, France), Hydrocortizone (500 ng/ml), cholera toxin (100 ng/ml), Insulin (10 μg/ml) (Sigma Aldrich, Saint-Quentin Fallavier, France) and antibiotics/glutamine (Invitrogen).

Human Fibroblasts were grown in MEM supplemented with 10% FCS, antibiotics/glutamine (Invitrogen).

## FACS and β-Galactosidase assays and cell culture

FACS and β-Galactosidase assays were performed as described in Jullien et al. (**Jullien et al., 2013**)

## Cell transfection and infection

Transient transfections were performed using TURBOFEC (Fermentas, Thermo Fisher) following the manufacturer's instructions. Cells were used at the indicated times or 24 hr after transfection.

Stable overexpressing clones were obtained by retroviral infection using pMXs-based constructs (Cell Biolabs, Euromedex, Mundolsheim, France). Briefly, amphotropic retroviruses were produced by transfection of the Phoenix cell line with the pMXs vector coding for the appropriate proteins using TURBOFEC. After two rounds of infection, cells were selected with puromycin (0.5 to 1.2 μg/ml, depending on the cell type) for three days.

Stable *GWL* knockdown cells were obtained by using pSIREN-RetroQ-based constructs (Clontech, Saint-Germain-en-Laye, France) and infected as above. The anti-*GWL* shRNA sequence corresponds to sequence 1 from Burgess et al. (**Burgess et al., 2010**).

The Rictor shRNA sequence (CCCCCCGCATTGTCTCTATCAAGTTCAAGAGACTTGATAGAGAC-AATGCGGTTTTTA) was cloned in pSR-GFP and cells were infected as described above.

## Generation of human fibroblast cell lines

293T producer cell line were transfected with retroviral pBabe-hTERT, pBabe-V12Hras or pZIP-SVX-SV40T constructs and VsVg and Gag/Pol vectors using Jet-Pei reagent. The resulting supernatants were used to infect foreskin fibroblasts. Retroviral construct were introduced serially; drug selection was used to purify cell population between infections. Cells were selected in G418 (400 μg/ml, 10 days), hygromycin (100 μg/ml, 7 days) or puromycin (1 μg/ml, 2–3 days) respectively. Retroviral vectors carrying only drug resistance genes were used as controls.

## Human phospho-kinase antibody array

The analysis of the phosphorylation profiles of kinases and their protein substrates in the different control and GWL overexpressed or knockdown cells were performed by Proteome Profiler Human Phospho-Kinase Array Kit (R&D Systems, Lille, France) as manufacture's instructions. In brief, cell lysate samples were diluted and mixed overnight with a cocktail of biotinylated detection antibodies. The sample/antibody mixture was then incubated with the array. Any protein/detection antibody complex present is bound by its cognate immobilised capture antibody on the membrane. Streptavidin-Horseradish Peroxidase and chemiluminescent detection reagents are added, and a signal is produced in proportion to the amount of protein bound. Chemiluminescence is detected in the same manner as a Western blot.

## In vitro phosphorylation assays

A baculovirus encoding human K72M mutant of GWL fused to GST was constructed by standard methods (Bac-to-Bac, GIBCO, Thermo Scientific) and inserted to baculovirus. Extracts of SF9 cells were collected after infection with the virus and the fusion protein was purified over glutathione-Sepharose beads (Pharmacia) by standard methods.

1 µg of recombinant AKT protein (Millipore, #14–279, Molsheim, France) was then mixed or not with 500 ng of recombinant GST-GWL (K72M) protein in a phosphorylation buffer (TRIS 50 mM, $MgCl_2$ 10 mM, ATP 100 µM) and 2 µCi of $ATP\gamma^{33}$. Thirty minutes later reactions were stopped by adding Laemmli sample buffer and analysed by SDS-PAGE. The phosphorylation of AKT or, as a positive control, 1 µg His-ARPP19 by GWL was subsequently determined by autoradiography. The S62A mutant form of ARPP19 was used as a negative control.

When phosphorylation of PHLPP by GWL was tested, Hek-293t cells were transfected with pcDNA3-HA-PHLPP plasmid and 24 hr later cell lysate was recovered. After 10-minutes centrifugation at 15 000 g, the supernatant was used for immunoprecipitation. A volume of the supernatant containing 700 µg of protein was mixed with 20 µl of magnetic Protein G-Dynabeads (Dynal, Thermo Fisher) pre-linked with 5 µl of anti-HA antibodies and incubated for 30 min at room temperature. Beads were washed three times with NETN buffer (20 mM Tris pH 8, 1 mM EDTA, 150 mM NaCl, 0,25% IGEPAL). Two different immunoprecipitates were then mixed with 10 µl of phosphorylation buffer and 2 µCi of $ATP\gamma^{33}$ and 500 ng of purified recombinant GST-GWL (K72M) protein. We used GST-ENSA protein as a positive control. Thirty minutes later reactions were stopped by adding Laemmli sample buffer and analysed by SDS-PAGE. The phosphorylation of PHLPP or GST-ENSA by GWL was subsequently determined by autoradiography.

When the activity of the different GWL forms was measured in the corresponding stable cell line, a volume of cell extract corresponding to 400 µg of total protein was used. Immunoprecipitation was then performed as for PHLPP in vitro phosphorylation assays except that anti-GWL antibodies instead of anti-HA were used. 1/5 of the immunoprecipitate was used to measure the levels of the different GWL forms and the rest was used for in vitro phosphorylation as for PHLPP using His-ARPP19 protein as a substrate.

## Cell invasion, migration and proliferation

Invasion of MDA-MB-231, D3H2LN and NIH3T3 cells was tested as previously described )***Sanz-Moreno et al., 2008***). Briefly, $10^5$ cells/ml were suspended in 2 mg/ml of serum-free liquid rat-tail collagen I (Becton Dickinson, Le Pont de Claix, France) and 100 µl/well were dispensed in black 96-well ViewPlates (PerkinElmer) coated with 0.2% BSA in DMEM supplemented with 1 µl of Fluorobeads (Molecular Probe, Thermo Fisher). Plates were centrifuged at 300 *g for 5 min* and incubated at 37°C/5% $CO_2$. After collagen polymerisation, 2% foetal calf serum (FCS) was added on top of the collagen. Forty-eight hours later, cells were fixed with 20% paraformaldehyde in PBS supplemented with 1 µg/ml Hoechst. Plates were then scanned in a Cellomics ArrayScan VTI HCS Reader (Thermo Scientific, Thermo Fisher) and cells located at 25–50 µm from the bottom were counted.

Migration of MDA-MB-231, D3H2LN and NIH3T3 cells was tested by using wound-healing assays. Cells were seeded in 35 mm culture-insert microdishes (Ibidi, Nanterre, France) until confluence. Inserts were then removed and cells were followed by time-lapse microscopy (Invers1 Zeiss Axiovert 200M) for 24 hr. Bright-field images were taken every 30 min. The area without cells between the cell surfaces was calculated for each frame and the covered surface between the initial and the

current frame was quantified and represented. Finally, the trendline equation was extrapolated and the covered surface (μm$^2$) per minute calculated.

For SW480 and SW620 cells, migration and invasion assays were performed in Boyden chambers (BD Bioscience) using 80,000 cells in the presence of 100 μL Matrigel (0.5 mg/mL; BD Bioscience) for invasion assays. Cells were fixed and stained 24 hr later in 4% paraformaldehyde/PBS with 1 μg/ml Hoechst and the number of cells present in the lower site of the filter counted with an Inverse1 Zeiss Axiovert 200M microscope.

For cell proliferation assays, Hek-293t, NIH3T3 and HeLa cells were seeded in 35-mm 6-well plates (10$^5$ cells/well). Cells were then counted with an Automated Cell Counter (Countess, Invitrogen, Thermo Fisher).

For cell proliferation assays, SW480, SW620, MDA-MB-231, D3H2LN, MCF10A cells and human fibroblasts were seeded at an initial density of 4000 cells per dish in 96-well dishes. At the indicated time points, cells were stained with crystal violet and the DNA content of attached cells was determined by measuring the absorbance at 595 nm.

To inhibit AKT during cell proliferation, invasion and migration assays, the AKT Inhibitor VIII (Millipore-Calbiochem) or MK2206 (Santa Cruz Biotechnology, Heidelberg, Germany ) was added to the culture medium 4 hr after plating cells and was replaced every two days. AKT phosphorylation on S473 was measured 24 hr after inhibitor treatment. Cell proliferation was measured in cells containing AKT inhibitor at the indicated times. Cell migration and invasion in SW480 and MDA-MB-231 was measured 24 hr after inhibitor addition. Cell invasion in MDA-MB-231 was recorded 48 hr after inhibitor addition.

## Cell contact inhibition

NIH3T3 and MCF10A cells overexpressing WT or mutant GWL were seeded in 60 mm plates and maintained in culture medium supplemented with 5% foetal bovine serum (NIH3T3 cells) or 2% horse serum (MCF10A cells) for 2 to 3 weeks until foci were evident. Cells were then fixed and stained with a 70% ethanol/1% crystal violet solution for 5 min, washed and colonies counted.

## Anchorage-independent growth and in vivo tumour formation

Growth of cells in soft agar was performed as previously described (***Hahn et al., 1999***).

For mouse experiments, 10$^6$ MDA-MB-231 cells overexpressing or not GWL or D3H2LN cells expressing control or *GWL* shRNA were injected subcutaneously in the right (control pMXs or shLUC) and left (GWL overexpressing or *GWL* shRNA) flank of athymic mice. Tumour formation was evaluated at the indicated times by palpation, and the size of the palpable tumours was measured with precision instruments.

All animal experiments conformed to the relevant regulatory standards and were approved by the Ethics Local Committee of the IRCM (Institut de Recherche en Cancérologie de Montpellier) and the Regional Ethic Committee of the "Languedoc Roussillon" (France) (Ref: 1137).

## Selection of normal and colorectal tumour samples

Forty patients with colorectal cancer and synchronous and unresectable liver metastases were enrolled in a prospective study at the ICM (Institut du Cancer de Montpellier) Cancer centre (from January 2000 to June 2004 (***Del Rio et al., 2007***). Normal colon, colon cancer and hepatic metastasis samples were collected at the time of surgery, prior to chemotherapy. Normal and paired cancer tissues from thirteen patients were used in this study.

The study was approved by the ICM CORT (Translational Research Committee) ethical committee and all participating patients were informed of the study and provided their signed written informed consent before enrolment. This set of samples were already used in Del Rio et al. (***Del Rio et al., 2013***)

## TMA construction and immunostaining

Cases were issued from the archives of the Department of Pathology of Bergonie Institute (Bordeaux, France). For all samples, ethical approval was obtained from the appropriate committees. Cases were then centralised in the Biological Resources Centre of Bergonie Institute, which has received the agreement from the French authorities to deliver samples for scientific research (AC-

2008-12). TMA was made from tissue biopsies of 80 patients with primary colorectal carcinoma operated and treated at the Bergonie Institute (Bordeaux, France) between January 1, 1999, and December 31, 2004. Patients and tumours characteristics are detailed in Soubeyran et al. (**Soubeyran et al., 2011**). Only 76 of the 80 patients of this TMA were included in this study due to loss of the normal tissue samples of four patients.

For each patient, a piece of tumour and normal mucosa were collected, fixed in Holland Bouin's fluid and stored as paraffin-embedded tissue samples. For TMA construction, 0.6 mm diameter tissue cores were extracted from these blocks and assembled in the recipient paraffin block using a tissue arrayer (Beecher Instruments Tissue Arrayer, NJ). The array was designed to contain three cores of tumour tissue placed alongside a core of matched healthy mucosa. Once completed, 5 µm sections were cut from the array, placed on glass slides and deparaffinised in toluene. After heat–induced proteolytic epitope retrieval in Tris-EDTA buffer (pH 9.0), slides were incubated with an anti-human GWL primary antibody at room temperature (RT) for 1 hr, washed in PBS thrice and then incubated with an Alexa Fluor 488 goat anti-rabbit secondary antibody (Molecular Probes) at RT for 1 hr. Nuclei were stained with 1 µg/mL DAPI solution in the dark for 15 min. Slides were finally dehydrated and mounted in Fluoromount–G. Image acquisition and quantitative analyses were performed using the Icys Research Imaging cytometer (Compucyte, MA) (**Henriksen et al., 2011**; **Pozarowski et al., 2013**). Once acquired, fluorescent signals were quantified by applying a phantom (=pseudocells) segmentation. GWL fluorescent signal was evaluated in areas delimited for each spot by DAPI staining and plotted as histograms, from which median intensity values were extracted. These median intensities were then reported in a scatter dot plot to compare GWL expression level in normal and tumour tissue. Data were computed and analysed using Excel and GraphPad.

## Plasmids and antibodies

Plasmids and antibodies are specified in **Supplementary files 1, 2**.

# Acknowledgements

We thank Dr. A Sirvent for the generous gift of the pSR-shRictor; V. Georget and S De Rossi from Montpellier RIO Imaging for microscopy facilities; C Vincent from the RAM animal facility; N Pirot and F Bernex from the RHEM histology facility and I Mahouche for TMA analysis. We also thank P Richard from the Antibody Production Platform at the CRBM. We are indebted to S Roche, P Roux and C Sardet for their helpful discussions. This work was supported by the Ligue Nationale Contre le Cancer (Equipe Labellisée) and the "Agence Nationale de la Recherche" (Programme Blanc). J Vera is a "Fondation pour la Recherche Medicale" fellow.

# Additional information

## Funding


| Funder | Grant reference number | Author |
|---|---|---|
| Agence Nationale de la Recherche | ANR-10-BLAN-1207 | Anna Castro |
| Fondation pour la Recherche Médicale | SPF20111223514 | Jorge Vera |
| Ligue Nationale Contre le Cancer | LNCC LORCA 2014 | Thierry Lorca |

The funders had no role in study design, data collection and interpretation, or the decision to submit the work for publication.


## Author contributions

JV, VG, TL, Conception and design, Acquisition of data, Analysis and interpretation of data, Drafting or revising the article; LL, Acquisition of data, Analysis and interpretation of data, Drafting or revising the article; SV, Conception and design, Acquisition of data, Drafting or revising the article; GG, AC, Conception and design, Analysis and interpretation of data, Drafting or revising the article; MDR, IS,

FC, Analysis and interpretation of data, Drafting or revising the article, Contributed unpublished essential data or reagents

### Ethics

Human subjects: Human Samples for TMA: Cases were issued from the archives of the Department of Pathology of Bergonie Institute (Bordeaux, France). For all samples, ethical approval was obtained from the appropriate committees. Cases were then centralised in the Biological Resources Center of Bergonie Institute, which has received the agreement from the French authorities to deliver samples for scientific research (AC-2008-812)." The TMA used in this study has been moreover used in three other published studies (See: Soubeyran and al., Am J of Pathology 2011, Rey and al., Cell Cycle 2013 and Oncogene 2015). For colorectal tumour samples used for western blot: The study was approved by the ICM CORT (Translational Research Committee) ethical committee and all participating patients were informed of the study and provided their signed written informed consent before enrolment. This set of samples were already used in Del Rio et al., JCO 2007 and Del Rio et al, Plos one 2013.
Animal experimentation: All animal experiments conformed to the relevant regulatory standards and were approved by the Ethics Local Committee of the IRCM (Institut de Recherche en Cancérologie de Montpellier) and by the Regional Ethic Committe of the "Languedoc Roussillon"(France). Ref: 1137.

## Additional files

### Supplementary files

• Supplementary file 1. Antibodies used in the study.

• Supplementary file 2. Plasmids used in the study.

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
