## [Decision Letter]

Thank you for submitting your work entitled "Greatwall promotes cell transformation, proliferation and invasion by preventing PHLPP-mediated Akt dephosphorylation" for consideration by *eLife*. Your article has been favorably reviewed by two peer reviewers, and the evaluation has been overseen by a Reviewing Editor and Charles Sawyers as the Senior Editor.

The reviewers have discussed the reviews with one another and the Reviewing editor has drafted this decision to help you prepare a revised submission.

Summary:

This paper investigates the role of the mitosis-regulatory kinase Greatwall (GWL) in tumorigenesis. Overexpression of GWL is shown to stimulate the proliferation of different immortalized cell lines and to induce cell transformation as well as enhanced migration and invasion. Conversely, GWL downregulation impairs the tumorigenic properties of two cancer cell lines. Importantly, the tumor-promoting activity of GWL appears to be independent of ARPP19 and ENSA, two key substrates of GWL in the regulation of mitosis, but instead correlates with the phosphorylation of AKT on Ser473. The authors present a number of experiments supporting that GWL tumorigenic effects are mediated by AKT and that GWL-induced degradation of the phosphatase PHLPP may account for its ability to activate AKT. GWL is also shown to be overexpressed in some colorectal tumors.

This is a very interesting paper illustrating a new function for GWL, which is mostly based on good quality data. The authors also present evidence for the upregulation of GWL in human tumors consistent with the pro-tumorigenic role proposed here.

Essential revisions:

1) Figure 1–Figure 4 are repetitive and can easily be reduced to 2 illustrations. Some parts of the illustrations could be moved to the supplementary section. The text should be reduced accordingly.

2) Figure 5 is confusing and can be dropped since only the AKT pathway is further explored. It suffices to mention that an altered AKT-S473 phosphorylation came out of an antibody array.

3) Figure 6—figure supplement 4: The MEK inhibitor U0126 is normally used at concentrations of 10-50 uM; it seems unlikely that 10 nM U0126 suffices to inhibit the ERK1/2 pathway. Unless the inhibition is confirmed, it is not correct to conclude that MEK inhibition does not affect the proliferation of GWL overexpressing 231 cells.

4) Figure 7—figure supplement 1, testing the possible direct phosphorylation of AKT by GWL. This is an important experiment but the negative result presented is not convincing. The Coomassie staining shows that there is much less AKT protein than the ARPP19 positive control. The in vitro phosphorylation should be repeated using the same amount of AKT and ARPP19 proteins, preferably over a range of concentrations to rule the possibility that AKT could be phosphorylated by GWL less efficiently than ARPP19. This would still not completely rule out a direct phosphorylation, but it would certainly make the evidence more convincing.

[Editors' note: further revisions were requested prior to acceptance, as described below.]

Thank you for resubmitting your work entitled "Greatwall promotes cell transformation by hyperactivating AKT in human malignancies" for further consideration at *eLife*. Your revised article has been favorably evaluated by Charles Sawyers (Senior Editor), a Reviewing Editor, and two reviewers. The manuscript has been improved but there are some remaining very minor issues that need to be addressed before acceptance, as outlined below:

1) V12-Ras in Figure 1 is not mentioned in the text.

2) Authors indicate in Figure 4—figure supplement 4 that inhibition of MEK was confirmed by decreased phosphorylation of Erk1/2 on S202/S204. This should be pErk1/2 (T202/Y204) antibody, which should be corrected in the figure and legend. The source of the pERk1/2 antibody used should be also indicated in [Supplementary-material SD1-data].

---

## [Author Response]

*Essential revisions: 1) Figures. 1-4 are repetitive and can easily be reduced to 2 illustrations. Some parts of the illustrations could be moved to the supplementary section. The text should be reduced accordingly.*

Figure 1–Figure 4 have been reduced to Figure 1, Figure 2. Results of the effect of transient overexpression of GWL in Hela and Hek293T cells have been eliminated from the manuscript. Data on cell proliferation, invasion and migration and cell contact inhibition in NIH3T3 cells overexpressing GWL have been moved to Figure 1—figure supplement 2. Finally, the results on cell proliferation, invasion and migration in SW480 overexpressing the different GWL forms and in SW620 knocked down or not of GWL were moved to Figure 2—figure supplement 2.

*2) Figure 5 is confusing and can be dropped since only the AKT pathway is further explored. It suffices to mention that an altered AKT-S473 phosphorylation came out of an antibody array.*

Figure 5 of the first version of the manuscript (now Figure 3) has been reduced. Only differences in the phosphorylation of GSK3 and AKT are shown.

*3) Figure 6—figure supplement 4: The MEK inhibitor U0126 is normally used at concentrations of 10-50 uM; it seems unlikely that 10 nM U0126 suffices to inhibit the ERK1/2 pathway. Unless the inhibition is confirmed, it is not correct to conclude that MEK inhibition does not affect the proliferation of GWL overexpressing 231 cells.*

In Figure 6—figure supplement 4 of the first version of the manuscript (now Figure 4—figure supplement 4) western blot data showing the phosphorylation of AKT and of ERK1/2 have been added. The results of this western blot show that pERK1/2 is higher in GWL overexpressing cells (as indicated in Figure 5–phosphokinase array of the first version of the manuscript) and that this dephosphorylation decreases in the presence of 10nM U0126 indicating that, at this dose, MEK is inhibited in GWL overexpressing cells. Moreover, these results indicate that pAKT on S473 does not decrease after U0126 addition demonstrating that the MEK-MAPK pathway is not involved in the hyperphosphorylation of AKT in GWL overexpressing cells.

*4) Figure 7—figure supplement 1, testing the possible direct phosphorylation of AKT by GWL. This is an important experiment but the negative result presented is not convincing. The Coomassie staining shows that there is much less AKT protein than the ARPP19 positive control. The in vitro phosphorylation should be repeated using the same amount of AKT and ARPP19 proteins, preferably over a range of concentrations to rule the possibility that AKT could be phosphorylated by GWL less efficiently than ARPP19. This would still not completely rule out a direct phosphorylation, but it would certainly make the evidence more convincing.*

In Figure 7—figure supplement 1 of the first version of the manuscript (now Figure 5—figure supplement 1) the in vitro phosphorylation has been repeated using the same amounts of AKT and ARRP19.

[Editors' note: further revisions were requested prior to acceptance, as described below.]

*The manuscript has been improved but there are some remaining very minor issues that need to be addressed before acceptance, as outlined below: 1) V12-Ras in Figure 1 is not mentioned in the text.*

The results showed in Figure 1 for V12Ras have been discussed in the main text (subsection “GWL overexpression promotes transformation of immortalized mammary gland cells and primary human fibroblasts”).

*2) Authors indicate in Figure 4—figure supplement 4 that inhibition of MEK was confirmed by decreased phosphorylation of Erk1/2 on S202/S204. This should be pErk1/2 (T202/Y204) antibody, which should be corrected in the figure and legend. The source of the pERk1/2 antibody used should be also indicated in [Supplementary-material SD1-data].*

“pErk 1/2 (S202/S204)” has been modified to “pErk 1/2 (T202/Y204)” in Figure 4—figure supplement 4 and in the corresponding figure legend. In addition, the reference for anti-pErk 1/2 (T202/Y204) antibody has been added into [Supplementary-material SD1-data].